# Development of an Electronic Stethoscope and a Classification Algorithm for Cardiopulmonary Sounds [note 1]

**DOI:** 10.3390/s22114263

**Published:** 2022-06-03

**Authors:** Yu-Chi Wu, Chin-Chuan Han, Chao-Shu Chang, Fu-Lin Chang, Shi-Feng Chen, Tsu-Yi Shieh, Hsian-Min Chen, Jin-Yuan Lin

**Affiliations:** 1Department of Electrical Engineering, National United University, Miaoli City 36003, Taiwan; a0976335098@gmail.com (F.-L.C.); sisterw961o3y94rm4@gmail.com (S.-F.C.); yuan@nuu.edu.tw (J.-Y.L.); 2Department of Computer Science and Information Engineering, National United University, Miaoli City 36003, Taiwan; cchan@nuu.edu.tw; 3Department of Information Management, National United University, Miaoli City 36003, Taiwan; cschang@nuu.edu.tw; 4Section of Clinical Training, Department of Medical Education, Taichung Veterans General Hospital, Taichung City 40705, Taiwan; zuyihsieh@gmail.com; 5Division of Allergy, Immunology and Rheumatology, Taichung Veterans General Hospital, Taichung City 40705, Taiwan; 6Center for Quantitative Imaging in Medicine (CQUIM), Department of Medical Research, Taichung Veterans General Hospital, Taichung City 40705, Taiwan; hsmin6511@gmail.com

**Keywords:** electronic stethoscope, cardiopulmonary sound classification, principal component analysis, Mel-frequency cepstral coefficients, ensemble learning

## Abstract

With conventional stethoscopes, the auscultation results may vary from one doctor to another due to a decline in his/her hearing ability with age or his/her different professional training, and the problematic cardiopulmonary sound cannot be recorded for analysis. In this paper, to resolve the above-mentioned issues, an electronic stethoscope was developed consisting of a traditional stethoscope with a condenser microphone embedded in the head to collect cardiopulmonary sounds and an AI-based classifier for cardiopulmonary sounds was proposed. Different deployments of the microphone in the stethoscope head with amplification and filter circuits were explored and analyzed using fast Fourier transform (FFT) to evaluate the effects of noise reduction. After testing, the microphone placed in the stethoscope head surrounded by cork is found to have better noise reduction. For classifying normal (healthy) and abnormal (pathological) cardiopulmonary sounds, each sample of cardiopulmonary sound is first segmented into several small frames and then a principal component analysis is performed on each small frame. The difference signal is obtained by subtracting PCA from the original signal. MFCC (Mel-frequency cepstral coefficients) and statistics are used for feature extraction based on the difference signal, and ensemble learning is used as the classifier. The final results are determined by voting based on the classification results of each small frame. After the testing, two distinct classifiers, one for heart sounds and one for lung sounds, are proposed. The best voting for heart sounds falls at 5–45% and the best voting for lung sounds falls at 5–65%. The best accuracy of 86.9%, sensitivity of 81.9%, specificity of 91.8%, and F1 score of 86.1% are obtained for heart sounds using 2 s frame segmentation with a 20% overlap, whereas the best accuracy of 73.3%, sensitivity of 66.7%, specificity of 80%, and F1 score of 71.5% are yielded for lung sounds using 5 s frame segmentation with a 50% overlap.

## 1. Introduction

Stethoscopes are an extremely important diagnostic tool in the medical community. The stethoscope can be used to listen to not only heart sounds to determine heart-related diseases but also lung sounds to diagnose whether there are abnormalities in the lungs. This is known as auscultation. Traditional stethoscopes utilized horn-shaped stethoscope heads for listening to the sounds of the movements of the visceral organs. However, with conventional stethoscopes, the auscultation results may vary from one doctor to another due to a decline in his/her hearing ability with age or his/her different professional training background, and the problematic cardiopulmonary sound cannot be recorded for further analysis. Therefore, electronic stethoscopes that can record and analyze/classify cardiopulmonary sounds are needed to cope with these issues. If an effective classification algorithm can be embedded into electronic stethoscopes, doctors can use electronic stethoscopes to obtain a prompt preliminary diagnosis in an emergency, or those who need secondary prevention after discharge from the hospital and care at home can have long-term monitoring and early detection of abnormalities.

Based on the sensor used in electronic stethoscopes, air- and contact-conduction electronic stethoscopes are two common types of electronic stethoscopes. The air-conduction electronic stethoscope utilizes an electromagnetic coil or electret capacitor as the sensor to collect sound signals with high stability and strong reliability but with low sensitivity. The contact conduction electronic stethoscope mostly uses piezoelectric materials as a sound sensor with improved anti-interference ability and sensitivity. However, any deformation or damage to the piezoelectric materials in the manufacturing process degrades the sensitivity of the stethoscope.

In the category of air-conduction electronic stethoscopes, McLane et al. [1] developed an advanced air-conduction electronic stethoscope using a microphone array, an external-facing microphone, and an onboard signal processor to perform adaptive noise suppression of lung auscultation in noisy clinical settings. Zhang et al. [2] designed an electronic auscultation system for the graphic recording of heart, lung, and trachea (HLT) sounds by placing microphones in a CNC-machined Delrin housing case covered by diaphragms. Sixteen acoustic sensors, of which 14 were positioned in a memory foam pad and 2 were placed directly on the heart and trachea, were used to record the acoustic data using a LabView program, and the waveforms in the time and frequency domain, as well as a spectrogram for visual examination, could be plotted using Matlab.

In the category of contact-conduction electronic stethoscopes, Toda and Thompson [3] devised a contact-vibration sensor by bonding a piezoelectric polyvinylidene fluoride (PVDF) film to a curved rubber piece having a front-contact face. Vibrations transmitted from the front-contact face through the rubber to the film cause pressure normal to the surface of the film and then an electric field is induced by the piezoelectric effect. Duan et al. [4] proposed a double-sided diaphragm micro-electro-mechanical system (MEMS) electronic stethoscope (DMES) based on a bionic lollipop-shaped MEMS sound-sensitive sensor. Shi et al. [5] designed a stethoscope also based on the bionic lollipop-shaped MEMS and developed an acquisition circuit and PC upper machine for real-time acquisition of heart sound signals. Recently, forcecardiography (FCG), a non-invasive technique that measures vibrations via force-sensing resistors (FSR), has been proposed [6,7] to measure heart mechanical vibrations. The active area of FSR was fixed with a rigid dome and an accelerometer was used to acquire the dorso-ventral seismocardiography (SCG) signal. In [6], the FCG sensor and the accelerometer were firmly mounted on a plexiglass rigid board. In [7], a lead-zirconate-titanate piezoelectric disk equipped with the same dome-shaped mechanical coupler used for the FSR-based sensor was proposed for simultaneous monitoring of respiration, infrasonic cardiac vibrations, and heart sounds.

Three steps are involved in the analysis or classification of auscultatory sounds: pre-processing, feature extraction, and classifier design. Pre-processing deals with the noise-processing or signal-frame decomposition. For noise processing, noise reduction and de-noising of the samples are performed to enhance the noise immunity of the classification algorithm. For cardiopulmonary signals, the location of the recorded cardiopulmonary sounds may vary but usually, both cardiac and pulmonary signals are included. Therefore, Chien et al. [8] used two microphones to collect signals from the left and right chest and presented a fast independent component analysis (ICA) algorithm to separate heart and lung sounds. Hadjileontiadis and Panas [9] and Liu et al. [8] used wavelet transform techniques to denoise heart sounds from lung signals [9] and, conversely, filter out lung sounds from heart sounds [10]. Mayorga et al. [11] proposed an empirical mode decomposition (EMD) followed by Gaussian mixed models (GMM) to improve ICA for separating heart and lung sounds. In addition to noise reduction, EMD is also a method to estimate the frequency components [12,13,14] and each component can be used as a classification feature. In [12], EMD was used to separate heart sounds from lung signals; in [13], EMD was used to capture heart murmurs; and in [14], EMD was used to segment and capture the underlying heart sounds (S1, S2). Varghees and Ramachandran [15] proposed an empirical wavelet transform (EWT)-based heart sound signal decomposition method by integrating both EMD and WT. Ntalampiras [16] transformed the lung sound signals to the frequency and wavelet domains before performing the subsequent analysis.

The pre-processed acoustic signals then go through feature extraction for time- and/or frequency-domain features. The main purpose of capturing time-domain features is to observe inter-beat or inter-respiratory variations. The frequency-domain features are most commonly expressed as coefficients of frequency spectrum or inverse frequency spectrum, and the signals are transformed to the frequency domain to observe the changes in the signals in the frequency domain. The Mel-frequency cepstral coefficient is widely used in sound processing [17,18]. Potes et al. [17] used frequency-domain features together with time-domain features. Chowdhury et al. [18] denoised and compressed phonocardiography (PCG) signals using a multi-resolution analysis based on the discrete wavelet transform (DWT), segmented PCG signals using the Shannon energy envelope and zero-crossing into four parts and extracted features from PCG signals using a Mel-scaled power spectrogram and Mel-frequency cepstral coefficients (MFCC). Kumar et al. [19] used a set of features: time, frequency, and statistical or phase space features.

The extracted features are finally fed into a designed classifier. In [17], a total of 124 time-frequency features were extracted from the PCG and input to a variant of the AdaBoost classifier, and PCG cardiac cycles decomposed into four frequency bands were input to a second classifier using a convolutional neural network (CNN). An ensemble of classifiers combining the outputs of AdaBoost and the CNN was designed to classify normal/abnormal heart sounds. In [18], a five-layer feed-forward deep neural network (DNN) model was used. In [19], a support vector machine (SVM) classifier was trained and applied for each of the feature sets. Li et al. [20] applied a wavelet scattering transform and multidimensional scaling method and then presented a twin SVM (TWSVM) to classify heart sound signals. Gjoreski et al. [21] utilized classic machine-learning (ML) to learn from expert features and end-to-end deep learning (DL) to learn from a spectro-temporal representation of the signal. Shuvo et al. [22] proposed a lightweight end-to-end convolutional recurrent neural network (CRNN) architecture for the automatic detection of five classes of cardiac auscultation using raw PCG signals. This model was tested on PhysioNet/CinC 2016 challenge dataset [23] achieving an accuracy of 86.57%.

In this study, a low-cost electronic stethoscope with noise reduction is proposed, and an effective classification algorithm, based on principal component analysis (PCA), MFCC, and ensemble learning, is developed for auscultatory sounds. A graphical user interface is established to save or replay the recorded sounds in Raspberry Pi. The presented system can be useful in medical care.

## 2. Materials and Methods

### 2.1. Design of Electronic Stethoscope

Figure 1 shows the system architecture of the designed electronic stethoscope. Two microphones are used, one is installed inside the stethoscope head for heart/lung sounds, and the other is optionally attached to the back of the stethoscope head for noise reduction. Sound signals are then amplified through the first amplifier circuits. Different filters are designed for heart and lung sounds. After the first filtering, the signals are amplified through the second amplifier circuits for the second filtering. The two processed sounds are then subtracted by a differential amplifier and transmitted to a Raspberry Pi through a 16-bit ADC with a sampling rate of 44.1 kHz.

A condenser microphone that could be easily found on the market at a reasonable price was utilized in this study. Condenser microphones without the magnet and coil generate voltage changes with changing distances between two diaphragms in the capacitor and present the advantages of being lightweight, small in size, and have high sensitivity, and they are often used in high-quality recording. Condenser microphones with higher sensitivity could record more sound details but would also easily absorb noise in the environment; therefore, they are more suitable for use in quiet studios. Figure 2a shows our design of the electronic stethoscope head [24] where the condenser microphone (mic 1) is placed. Figure 2b shows the head with mic 1 inside, the cork, and the diaphragm. Here, a round disk cork is presented and is used to cover the back of the head if desired. Figure 2c illustrates the edge of the head (face up) surrounded by the cork. No gasket is stuffed inside the head. Figure 2d shows the back of the head attached to a microphone (mic 2) to collect environmental noise. Figure 2e depicts the front of the head encased with the diaphragm; the edge of the head is surrounded by the cork. A shielded line was used to connect the microphone and amplifier circuits. Two-stage amplification/filtering is adopted to process the small signals collected by microphones and more accurate heart/lung sound waveforms are obtained when compared with only one-stage amplification/filtering. The frequency of the heart sounds is from 1 Hz to 800 Hz and the human ear is sensitive in the range of 40 Hz to 400 Hz s most of the signals below 20 Hz are inaudible [25]. Therefore, the filtering band of heart sounds in this study is below 400 Hz and the low-pass filter is designed to attenuate the signals above 400 Hz. The main frequency range of the lung sounds is from 100 Hz to 2000 Hz, and the high-pass filter and low-pass filter are designed to filter out signals below 100 Hz and above 2000 Hz. Therefore, after amplification, the heart or lung sound is filtered by the corresponding filter to filter out the unwanted frequencies of sound. As the two processed heart/lung signals and noise signals need subtraction, the circuit signal-processing delay would affect noise reduction. To solve this problem, two-stage amplification/filtering circuits with different gains were applied for noise sounds. Figure 3 and Figure 4 illustrate amplification circuits for heart/lung sounds and noise sounds, respectively. Figure 5 depicts the filter circuits for heart and lung sounds. The same filters were applied to noise sounds. Figure 6 shows the whole circuit diagram where amplifiers and filters are marked for better understanding.

A graphical user interface (GUI) for recording sounds was also developed during the course of this study. For noise reduction, we designed 5 different models of electronic stethoscope heads as shown in Table 1 [24], where a commercial electronic stethoscope was included. Two microphones were used in Models 1~2, one inside the stethoscope head for collecting heart/lung sounds and the other on the back of the stethoscope head for collecting environmental noise. The difference between these two models is that one is without a cork (Model 1) and the other one is surrounded by a cork (Model 2, shown as in Figure 2c,e). Two stethoscope heads connected back-to-back are used in Models 3 and 4 and each stethoscope has its own microphone inside the head. The difference between these two models is that one is without a cork (Model 3) and the other one is surrounded by a cork (Model 4) as in Model 2. The difference between Models 1–2 and Models 3–4 is the way the microphone is used to collect environmental noise. Mic 2 in Models 1–2 is exposed to the air, whereas the one in Models 3–4 is inside the stethoscope head. Therefore, by comparing Models 1 and 3 (or Models 2 and 4), we can see the effects of noise reduction by the subtracter when a different way of collecting environmental noise is used. When compared with Model 1, Model 2 is designed to see the effects of noise isolation on the cork. When compared with Model 3, the design of Model 4 is also to see the effects of noise isolation on the cork. One stethoscope head (one microphone inside the head) fully covered by a cork is used in Model 5, which is different from Model 2 by missing one microphone on the back of the head and adding an additional piece of cork (as shown in Figure 2b) to cover the back of the head; therefore, the design of Model 5 is to see the effects of noise isolation on the cork without using the subtracter.

To test the noise reduction effect of the designed circuits, two experimental sites were set up. Figure 7 shows the schematic diagram of experimental site I in which a thick book is used for simulating the human chest wall. A regular speaker for PCs, a MECMAR speaker (Boxe PC Mecmar 2.0, 240 W PMPO) with a dimension of 70 mm (width) × 163 mm (height) × 70 mm (depth), was used. Ward construction noise [26] and airport noise [27] were selected for observation. The distance of the stethoscope head from the sound is about 1 m for the measurement. The audio sounds of noise received from the stethoscope head and the microphone and the audio sound after noise-canceling were analyzed by FFT for comparison. Figure 8 shows the experimental site II in which the stethoscope on a human body is tested. MECMAR speaker was utilized for playing the noise source. Ward construction noise [26] was selected as the noise source, the sound volume was adjusted to the maximal, and the distance of the measurement position is 1 m away from the speaker. The sound source [26] at 0:30~0:40 was measured for all testing. In order to isolate the interference of heart sounds, the skin of the lower leg, which is the farthest from the heart, was chosen as the medium for the measurement. The sound measurement (measurement 1) using the stethoscope on the skin of the lower leg was performed in a quiet environment. Then, the sound measurement on the skin of the lower leg was performed again in a noisy environment to collect both sounds from the stethoscope head (measurement 2) and from the microphone designed for collecting the environmental noise (measurement 3). The noise frequencies of these two sounds were compared. The measurements for heart sounds were conducted by placing the stethoscope head on the left chest (second intercostal space at the left edge of the sternum) in a quiet environment (measurement 4) and then in a noisy environment (measurements 5 and 6). When in a noisy environment, the measurements from the stethoscope without using the subtracter (measurement 5) and with using the subtracter (measurement 6) were compared by FFT.

### 2.2. Heart/Lung Sound Classification

In this subsection, we focus on the design of a heart/lung sound classification algorithm. In Taiwan, collecting patient data for analysis must first be approved by the Institutional Review Board (IRB). Moreover, even the approval is granted, collecting enough data is time- and labor-consuming. For the convenience of research without going through an Institutional Review Board (IRB) review, the study of the presented algorithm is focused on the heart and lung sound samples from the public domain [28,29]. These cardiopulmonary samples were first pre-processed by segmentation, dimensionality reduction, and signal processing, then the time- and frequency-domain features of the samples were extracted and the classifier model was designed by using ensemble learning, and finally, these feature vectors were fed into the classifier model for training and testing.

#### 2.2.1. Pre-Processing

In this study, the heartbeat frequency (50–80 beats per minute) and respiration frequency (12–20 beats per minute) were assumed for segmentation. To keep the complete information of one heartbeat or one respiration in a small frame, different lengths of small frames were tested. The heart and lung sound signals were segmented into small frames. Different lengths of small frames were tested to see which length would give the better performance, 1, 1.5, and 2 s for heart sounds, and 3, 4, and 5 s for lung sounds. In addition, overlapping ratios, such as 0%, 20%, and 50%, were adopted to segment the original samples in order to increase the number of samples and to see if the overlapping segmentation approach would give better recognition results. After segmentation, the small frame signals were subjected to principal component analysis (PCA) [30] and then the original signals were subtracted based on PCA to obtain the difference signals (frames after PCA). In our experience, abnormal heart sounds can be divided into three categories: heartbeat components, abnormal beating components, and noise. The heartbeat components refer to major cardiac cycles. For a normal or an abnormal heart sound, there exist these cardiac cycles: diastole, systole, diastole, and systole. For an abnormal heart sound in terms of PCG, there exist irregular vibrating waves during these cycles. These vibrations here refer to abnormal beating components. Basically, the heartbeat sound will account for more than 80% of the total input signal and these signals will often dominate the classification results. The noise or abnormal heart/lung sounds account for a small amount of the original signal. However, what we need and are concerned about are the abnormal beating sounds. Therefore, we use principal component analysis to remove the first 85–95% of the principal component to reduce the classification impact dominated by the heartbeat sound, leaving the signal data with abnormal sound as the main component for feature extraction. Figure 9 shows the sound frames before and after PCA for normal and abnormal heart sounds. Figure 10 shows the sound frames before and after PCA for normal and abnormal lung sounds. The horizontal axis represents the sampling points and the vertical axis represents the sound amplitudes in a wav format. In addition, the number of components to be removed can be decided according to the degree of contribution and usually, the principal components with a cumulative contribution of 85–95% are taken. In this study, 95% cumulative contribution was used.

#### 2.2.2. Feature Extraction

As mentioned in the previous section, although the difference signals can be directly observed by the naked eye, they are still too complex for computers. Therefore, the difference signals need to be characterized for easier identification. In this study, two kinds of features, time-domain features and frequency-domain features, were extracted.

First of all, in terms of time-domain features, the original difference signals are too high-dimensional and not easy to handle. With a sampling rate of 2000 Hz, a 2 s sound frame would have 2 × 60 × 2000 (240,000) points, which is a high-dimension vector. This high-dimension vector can be processed by MFCC and statistical feature extraction to reduce its dimension. Statistical time-domain features were used to effectively simplify the high-dimensional difference signals while retaining the characteristics of the original signals. In this study, 11 statistical features were selected: mean, standard deviation, mean absolute deviation, median, first quartile, third quartile, interquartile range, skewness, kurtosis, Shannon entropy, and spectral entropy.

Secondly, the frequency-domain information of sound signal processing is more likely to show differences than the time-domain features because the cardiopulmonary signals have periodic characteristics and different sounds have different frequencies. The Mel-frequency cepstral coefficients (MFCC) [31] are suitable frequency-domain features that are closer to the characteristics of the human ear in analyzing sound than the general spectrum or the inverse spectrum coefficients. The reason is that the human auditory system only responds linearly to frequencies below 1 KHz but rather shows a logarithmic function at higher frequencies. By using this relationship, the MFCC are spectral features and are obtained as follows. The sound signal is first pre-reinforced, such as passing the signal through a high-pass filter to enhance the information of a high frequency. This is because the energy of a high frequency is usually smaller than that of a low frequency. Then, Fourier transform is performed to obtain the power spectrum. The power spectrum obtained from each audio frame is then passed through a Mel filter to obtain the Mel scale. Forty Mel filters are usually used. Then, the logarithmic energy is extracted for each Mel scale, and discrete cosine conversion for the inverse spectrum domain is performed. Since the coefficients after filtering are highly correlated, the correlation can be removed and downscaled by discrete cosine conversion and the Mel-frequency inversion coefficient is the amplitude of the Mel-frequency inversion spectrum. In this study, 12 coefficients were used and the energy of the audio frame was superimposed to form the 13th coefficient. In addition, the maximum value in the power spectrum, the frequency of the maximum value in the power spectrum, and the percentage of the maximum energy in the power spectrum to the total energy were calculated. A total of 16 frequency-domain features together with 11 statistical features were used as the input feature vectors for the classifier. So, the dimension is reduced from 240,000 to 27 for a 2 s sound frame.

#### 2.2.3. Classifier

The classification algorithm is used to classify normal (healthy) and abnormal (pathological) heart sounds and normal and abnormal lung sounds. In this study, we adopted the concept of ensemble learning [32] to design the classifier model and experimented with several classical ensemble learning methods, including Bagging (bootstrap aggregating), AdaBoost (adaptive boosting), GentleBoost (gentle adaptive boosting), LogitBoost (adaptive logistic), and RUSBoost (random under-sampling boosting), to see which learning method is most suitable for the database selected for this study. Each sample segmentation approach (e.g., 0% overlap in 1 s, 20% overlap in 1 s, etc.) is trained 5 times, and an ensemble learning method is randomly selected for each time. The final classifier model is determined based on the best observation obtained by Bayesian optimization. The original sample was divided into many small frames for training and the model identified the results of the classification of the small frames for voting. If the number of abnormal frames exceeded a specific threshold, the original sample is regarded as an abnormal cardiopulmonary sample, whereas the opposite is a normal one. The experimental results are displayed according to different proportions (e.g., 5%, 10%, 15%, …, 95%) of the voting results and it is observed which proportion has the best voting results. In addition, if the length of an original sample could not be divided by more than one small frame, the sample is ignored and not counted.

To evaluate the performance of the algorithm proposed in this paper, four common evaluation metrics are used (Equations (1)–(5)), including accuracy (Acc), sensitivity (Se), specificity (Sp), and F1 score, which are commonly used to evaluate the performance of “abnormality recognition algorithms” (e.g., disease diagnosis). Accuracy is the most intuitive way to evaluate the average accuracy of the algorithm. Specificity focuses on the misdiagnosis, and higher specificity means lower misdiagnosis. Sensitivity evaluates the ability to detect patients, and higher sensitivity means better ability to identify patients. The F1 score is the summed average of the above scores and is used to summarize the overall performance of the algorithm.
(1)$$\mathrm{accuracy}=\frac{TP+TN}{TP+FN+FP+TN}$$
(2)$$\mathrm{specificity}=\frac{TN}{FP+TN}$$
(3)$$\mathrm{sensitivity}=\frac{TP}{TP+FN}$$
(4)$$\mathrm{precision}=\frac{TP}{TP+FP}$$
(5)$$F1\;\mathrm{Score}=\frac{2\times \mathrm{sensitivity}\times \mathrm{precision}}{\mathrm{sensitivity}+\mathrm{precision}}$$
where *TP* stands for true positive (with disease and classified as abnormal), *TN* stands for true negative (without disease and classified as normal), *FP* stands for false positive (without disease but classified as abnormal), *FN* stands for false negative (with disease but classified as normal).

## 3. Results

### 3.1. Design of Electronic Stethoscope

Figure 11a shows the PCB circuit of the presented system in accordance with the size of Raspberry Pi 3 Model B (85 mm × 55 mm). Figure 11b [24] shows the whole designed system. The gain of the first amplifier for the heart/lung sounds is designed at 1.0 so that the gain of the first amplifier for mic 2 (noise) does not need to be adjusted to a high gain. The gain of the second amplifier for the heart/lung sounds was adjusted to roughly 3.0. For the experimental site I (Figure 7), two different noise sources were played. Figure 12 shows the 10 s sound waveforms and FFTs measured by mic 1, mic 2, and the subtracter (noise reduction) for the ward construction noise that was played from the computer with a 40% volume and a speaker with the largest volume. Based on the FFT spectra, the noise amplitude peak before noise reduction is about 0.0013, whereas it is 0.0003 after noise reduction. Apparently, the volume after noise reduction is 4 times lower than before noise reduction; the noise reduction is about 12.74 dB based on the formula for the signal-to-noise ratio (SNR). The 10 s sound waves and FFTs measured by mic 1, mic2, and noise reduction for the airport noise are shown in Figure 13. For the airport noise played from the computer with the maximal volume and a speaker with the largest volume, the noise amplitude peak before noise reduction is about 0.0013 and about 0.00025 after noise reduction. In this case, the volume after noise reduction is 5.2 times lower than before noise reduction; the noise reduction is about 14.32 dB.

For the direct measurements of the human body in experimental site II (Figure 8), ward construction noise was used as the noise source and the test volume was adjusted to the maximal. The different models of stethoscopes listed in Table 1 were tested. For conciseness, 10 s waveforms and FFTs of measurements 1–6 only for Model 2 are shown in Figure 14. For Model 2, Figure 14a shows that the frequency band of the signals received from the contact of the stethoscope head with the skin of the lower leg is in 0~100 Hz. In comparison with Figure 14b, the frequency band of the noise received at the stethoscope end on the skin of the lower leg is in 150~450 Hz and the audio frequency peak appears at 300 Hz. Figure 14c reveals that the frequency band of the noise received by mic 2 at the ambient end distributes in 150~550 Hz and the audio frequency peak appears at 375 Hz. Figure 14d shows that the frequency band of the heart sound signals in a quiet environment is in 0~150 Hz and is mixed with the noise of the skin. Figure 14e reveals that the frequency band of the noise received at the stethoscope end is in 200~400 Hz and the audio frequency peak appears at 325 Hz. Figure 14f shows that the frequency band of the noise residual signals after noise reduction is in 250~450 Hz. Table 2 summarizes the observations of the tests.

Table 3 shows the FFT spectra of the heart sound signals with and without the subtracter, measured at the stethoscope end using four different models. Model 5 has no subtracter and Model 6 does not provide the option of using the subtracter. Therefore, Models 5 and 6 are not listed in Table 3. The results of noise reduction using the subtracter circuit for these four models are not satisfactory. However, when comparing the FFT spectra of the heart sounds obtained using the subtracter, Model 4 shows the lowest noise amplitude. When the subtracter is not used, the FFT spectra of the heart sound signals of the six Models (including the electronic stethoscope from Thinklabs One) are shown in Figure 15. Model 4 shows the lowest noise amplitude when measuring heart sounds in a noisy environment. Figure 15 also shows that the cork can reduce the noise by 1.9 dB between Model 1 and Model 2 and by 2.8 dB between Model 3 and Model 4. Model 4 can reduce the noise by 4.7 dB when compared to the commercial one, Model 6. 

### 3.2. Heart/Lung Sound Classification

Two sample databases were adopted from the Internet [28,29]. The database [28] contains 3240 heart sounds in wav format, including 2548 normal heart sounds and 692 abnormal heart sounds, with lengths of between 5 and 120 s and a sampling rate of 2000 Hz. The Lung Sound Dataset [29] is a collection of 920 lung sound recordings in wav format with lengths ranging from 10 to 90 s, created by two research teams in Portugal and Greece, containing 35 normal lung sounds, 1 asthma, 16 bronchiectasis, 13 occlusive bronchiolitis, 793 chronic obstructive pulmonary diseases (COPD), 37 pneumonia, 23 upper respiratory tract infection (URTI), and 2 lower respiratory tract infection (LRTI) sounds. The samples containing no noise from [28] were split into five folds and the samples for training and testing were randomly selected. The sample sizes for training and testing are shown in Table 4. In addition, samples containing noise in the database set were categorized as test samples (samples) to evaluate the tolerance of the proposed algorithm to noise. The score of each index was evaluated by averaging the 5-fold classification results. The classification results of the heart sound dataset with different segmentation lengths (1 s, 1.5 s, and 2 s), different overlap ratios (0%, 20%, and 50%), and different voting ratios (5–95%) are presented in Figure 16, Figure 17, Figure 18 and Figure 19 in terms of four metrics (accuracy, sensitivity, specificity, and F1 score).

Furthermore, sounds from the lung sound database [29] were used as the training and testing samples for the proposed algorithm. This database contains several types of lung sounds. However, the number of samples of some types is too small. Therefore, only 30 normal lung sounds and 30 abnormal lung sounds (including 6 bronchial dilatation, 6 occlusive bronchitis, 6 chronic obstructive pulmonary disease, 6 pneumonia, and 6 upper respiratory tract infection) were used to split the samples into three folds, and the samples were classified in a binary way, i.e., all lung-related diseases were considered as abnormal lung sounds. The number of training and test samples was 20 and 10, respectively. Similarly, the results were observed by four metrics: accuracy, sensitivity, specificity, and F1 score, as shown in Figure 20, Figure 21, Figure 22 and Figure 23, respectively.

## 4. Discussion

### 4.1. Design of Electronic Stethoscope

The presented stethoscope is cost-effective and easily implemented. The condenser microphone used in the presented stethoscope can be found on the market at a reasonable price and the filter and amplification circuits are simple. From the test results (Figure 12 and Figure 13) for experimental site I, the microphone at the noise-receiving end over-received the sound at low frequencies; therefore, a superposition of noise signals in the low-frequency section was observed. Even so, the noise reduction effect is still satisfactory, being 4~6 times lower than in the case not using the subtracter. However, this phenomenon affected the noise reduction when the stethoscope was applied to human skin. With the human body test for experimental site II, the noise reduction effect was far different from the test results conducted in experimental site I. It is because the noise peak frequency received at the stethoscope end on the skin of the lower leg (Measurement 2) is at 300 Hz, whereas the noise peak frequency received by mic 2 at the ambient end above the skin of the lower leg (Measurement 3) is at 350 Hz for Models 3 and 4 or 375 Hz for Models 1 and 2, as listed in Table 2. There is a frequency shift due to different sound media. The media for the microphone inside the stethoscope are the skin, diaphragm, and the air. The medium for the microphone of Models 1 and 2 to collect environmental noise is the air. The media for the microphone of Models 3 and 4 to collect environmental noise are the diaphragm and the air. When heart sounds were mixed with the noise, the noise peak frequency (Measurement 5) became 325 Hz for Models 1 and 2 or 310 Hz for Models 3 and 4 (see Table 2). This peak frequency shift resulted in some noise signals being superposed, as shown in Table 3, when the subtraction of two signals was performed. Nevertheless, when Models 1–4 all used the subtracter, the design of Model 4 outperformed the others (Models 1–3) with the lowest noise amplitudes. Even when the subtracter was not used for Models 1–4, Model 4 still performed better than the others. The cork surrounding the double heads in Model 4 presents certain noise isolation.

### 4.2. Heart/Lung Sound Classification

For the testing samples of the heart sounds in Table 4, we can see from Figure 16a that the best overall accuracy falls in the range of 84.3–86.9% at 20–40% of the voting ratios. The sensitivity and the specificity are inversely proportional to each other, with one being higher and the other lower. The sensitivity falls in the range of 42.2–88.3% and the specificity falls in the range of 76.6–98.8%. The trend shows that the sensitivity is greater than 76.6% only when the voting ratio is less than 25%. When the specificity is also considered, i.e., Figure 17a and Figure 18a are considered together, the best results are obtained at 20% and 25% of the voting ratios. The average sensitivity and the average specificity are 81.2% and 89.4%, respectively, at a voting ratio of 20%. The average sensitivity and the average specificity are 79.8% and 90.7%, respectively, at a voting ratio of 25%. The best overall F1 score is in the range of 83.1–86.2% at 10–20% of the voting ratios and the highest overall score was obtained at 20% of the voting ratio, as shown in Figure 19a. Based on the above four metrics (accuracy, sensitivity, specificity, and F1 score), the overall best score is at 20% of the voting ratio.

For the noisy samples of the heart sounds, the best overall accuracy is in the range of 25–80% of the voting ratios with a value of 66.1–70.4% as seen in Figure 16b. The sensitivity value ranges from 34.3 to 72.7% and the specificity value ranges from 58.5 to 92.7%. From the trend in Figure 17b, the sensitivity is greater than 58.5% only when the voting ratio is less than 25%, and furthermore, when considering the specificity of Figure 18b, the best performance occurs when the voting ratio is 25%, with an average sensitivity of 65.1% and an average specificity of 71.3%. From Figure 19b, the best overall F1 score is in the range of 63.6–69.5% at 5–40% of the voting ratios. Based on the four metrics, the best overall score is at 25% of the voting ratio.

The experimental results of different segmentation methods for heart sounds were compiled and discussed according to different voting ratios. Although nine different segmentation methods were presented, only the top three scoring methods are discussed here. From the results shown in Figure 16a, Figure 17a, Figure 18a and Figure 19a for the testing samples, the best accuracy was obtained with a 20% voting ratio and a segmentation length of 2 s, the best sensitivity was obtained with a segmentation length of 1 and 1.5 s and an overlap ratio of 50%, the best specificity was obtained with a segmentation length of 2 s and an overlap ratio of 50%, and the best F1 score was obtained with a segmentation length of 1 or 2 s and overlap ratios of 20% or 50%. Based on these four metrics, the model using the segmentation length of 2 s and overlap ratio of 20% achieves the best overall score. The scores are 86.9% for accuracy, which is slightly higher than that of [22], 81.9% for sensitivity, 91.8% for specificity, and 86.1% for F1 score. For further comparison considering the sample source, we chose the top three models [17,33,34] in the 2016 PhysioNet/CinC Heart Sound Classification Challenge [28]. Since the original training code of the authors participating in the 2016 competition was not available, only the final classification program and the trained weights of the neural network were available, so it was not possible to compare the results fairly. The comparison results are shown in the last four rows of Table 5. The results show that the recognition of normal heart sounds using our model is a little better than those using the other methods but the recognition of abnormal heart sounds is a little worse. Although the overall score was about 4–7% different from the top three in the competition, the classification method they used was a very complex DNN method that required special hardware with high computing power to execute and, in addition to the higher cost and inconvenience of porting the model, these methods could not be executed on embedded systems with low computing power. For outpatient physicians, they are unable to obtain the patient’s heart sound recognition results in real-time to assist them in making correct judgments. On the other hand, our model is not complicated and with reasonable performance, it can be imported into an embedded system such as Raspberry Pi. The performance indices of other models [17,18,20,21,22,33,34] that were reviewed in the Introduction are also listed in rows 1–7 of Table 5. However, since we don’t have access to the codes of these models, Table 5 only lists the test results collected from the literature.

From the results shown in Figure 16b, Figure 17b, Figure 18b and Figure 19b for noisy samples of heart sounds, at 25% of the voting ratio, segment lengths of 1.5 and 2 s and overlap ratios of 20% and 50% provide better performance in terms of accuracy; segment lengths of 1.5 and 2 s and overlap ratios of 20% offer better performance in terms of sensitivity; segment lengths of 1.5 and 2 s and overlap ratios of 50% give better performance in terms of specificity; and segment lengths of 1.5 and 2 s and overlap ratios of 20% and 50% offer better performance in terms of F1 score. Therefore, based on these four metrics, the model using a segmentation length of 2 s and an overlap ratio of 20% obtains the highest overall score: 69% for accuracy, 68.3% for sensitivity, 69.7% for specificity, and 68.8% for F1 score.

In summary, the signal segmentation length of 2 s and the overlap of 20% or 50% were found to be the most effective for the heart sounds. This indicates that a 2 s segmentation length can contain the most complete information for a single heartbeat. In addition, some degree of overlap is also effective in improving the overall classification accuracy.

As shown in Figure 20 for the lung sound classification, there is no significant change in the voting ratios for accuracy and the trend is similar. From Figure 21 and Figure 22, the sensitivity falls in the range of 53.3–70% and the specificity falls in the range of 50–93.3%. The voting ratio does not have much influence on these scores, only a slight upward or downward trend occurs at around 30–35% and 70–75% of the voting ratios. The best overall F1 score, as seen in Figure 23, falls at 56.7–71.5% at 5–30% of the voting ratios. Based on these four metrics, the best overall score occurs at 5–30% of the voting ratios. However, from the experimental results, the voting ratio did not have much effect on the scores and there was no significant change from 5% to 65% of the voting ratios. This might be caused by the high proportion of abnormal lung sound samples or uneven distribution. The segmentation length of 5 s with 50% overlap gave the best classification result: 73.3% for accuracy, 66.7% for sensitivity, 80% for specificity, and 71.5% for F1 score. This indicates that a segmentation length of 5 s can contain the most complete information about one respiration. In addition, allowing a certain degree of overlap in the segmentation as in the case of heart sounds, can effectively improve the overall classification accuracy.

In summary, the above results show that the voting ratio usually correlates with the metrics. There are three factors that affect the trends in the metrics. The first is the percentage of abnormal heart sounds or lung sounds in the samples. In medical terms, if a heart sound or lung sound is diagnosed as abnormal, not every heartbeat in that heart sound or respiration in that lung sound may be abnormal. The second is the total number of small frames segmented from a given sample. The higher the number of frames, the smoother the score curve and the easier it is to see the trend and the more accurate the results. The last one is the accuracy of the classifier in training the frames. The higher the accuracy of the frames, the higher the accuracy of the subsequent voting.

## 5. Conclusions

In this paper, the design of an electronic stethoscope and an AI classification algorithm for cardiopulmonary sounds were addressed. Five models of electronic stethoscopes have been proposed and tested. In Models 1 and 2, a microphone is installed inside the stethoscope head to collect heart and lung sounds and a second microphone is attached to the back of the stethoscope head for collecting environmental noise. In Models 3 and 4, double stethoscope heads where each has a microphone installed are connected back-to-back, one stethoscope head is for collecting heart and lung sounds and the other is for collecting environmental noise. Cork is used in Models 2 and 4 to isolate environmental sounds. In Model 5, only one stethoscope head covered by cork is used. The collected sounds are processed through two-stage amplification and filter circuits. Each processed heart/lung sound then optionally subtracts the processed noise sound for noise reduction and a Raspberry Pi is used to record the final sound. The effect of noise reduction for the presented electronic stethoscopes was tested in two different experimental sites. When subtraction is not used, Model 4 presents better performance with fewer noise amplitudes. When subtraction is used for Models 1–4, Model 4 still outperforms the other 3 Models with the lowest subtracted noise signals. The cork surrounding the double heads in Model 4 presents certain noise isolation. However, frequency shifts due to different sound media associated with the stethoscope head and the microphone for the noise were observed during the course of the study. Therefore, this issue may need further investigation.

For the cardiopulmonary sound classifications, a voting ensemble learning approach combined with PCA and MFCC was developed. Two public databases were used for training and testing. Different segmentation lengths (1 s, 1.5 s, and 2 s) and different overlap ratios (0%, 20%, and 50%) were applied to segment one sample into several small frames. Four common metrics (accuracy, sensitivity, specificity, and F1 score) were used to evaluate the performance of the algorithm. After testing, the best voting for heart sounds falls at 5–45% and the best voting for lung sounds falls at 5–65%. Based on the results for the heart sound testing samples containing no noise, the best overall score is obtained using 2 s frame segmentation with a 20% overlap: 86.9% for accuracy, 81.9% for sensitivity, 91.8% for specificity, and 86.1% for F1 score. For the lung sound testing samples, the best overall score is yielded using 5 s frame segmentation with a 50% overlap: 73.3% for accuracy, 66.7% for sensitivity, 80% for specificity, and 71.5% for F1 score. The signal segmentation length is long enough to cover one heartbeat or one respiration and a certain degree of overlap in segmentation would effectively improve the overall classification performance.

## Figures and Tables

**Figure 1 sensors-22-04263-f001:**
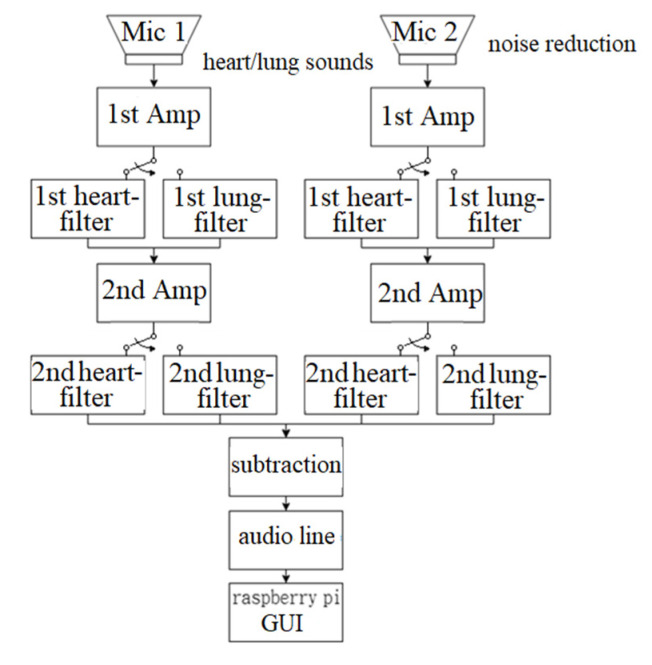
System architecture of electronic stethoscope.

**Figure 2 sensors-22-04263-f002:**
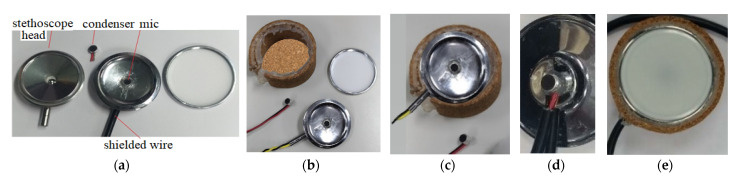
Condenser microphone placed in stethoscope. (**a**) a mic put into a stethoscope head, (**b**) a stethoscope head with mic 1 inside, the cork, and the diaphragm, (**c**) the edge of a head (face up) surrounded by the cork, (**d**) the back of the head attached to a microphone (mic 2), and (**e**) the front of the head encased with the diaphragm; the edge of the head surrounded by the cork

**Figure 3 sensors-22-04263-f003:**
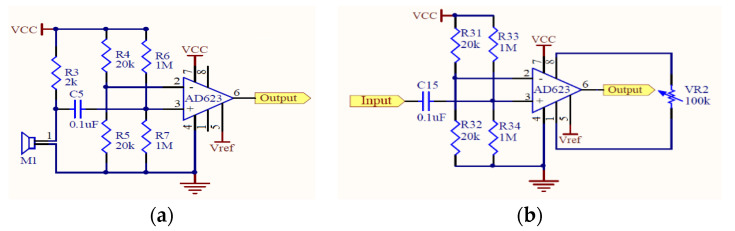
(**a**) First-stage amplification circuit and (**b**) second-stage amplification circuit for heart/lung sounds.

**Figure 4 sensors-22-04263-f004:**
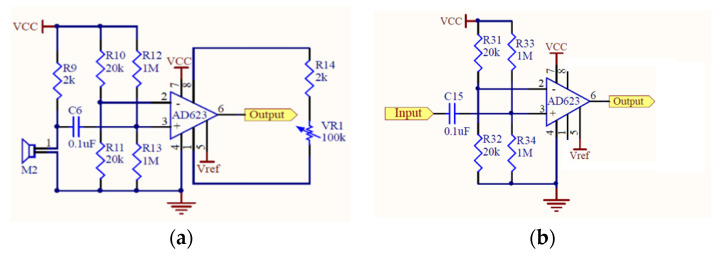
(**a**) First-stage amplification circuit and (**b**) second-stage amplification circuit for noise sounds.

**Figure 5 sensors-22-04263-f005:**
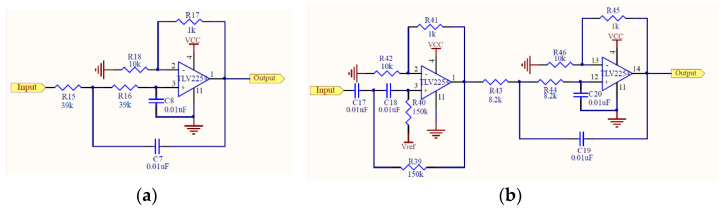
(**a**) Filter circuit for heart sounds and (**b**) filter circuit for lung sounds.

**Figure 6 sensors-22-04263-f006:**
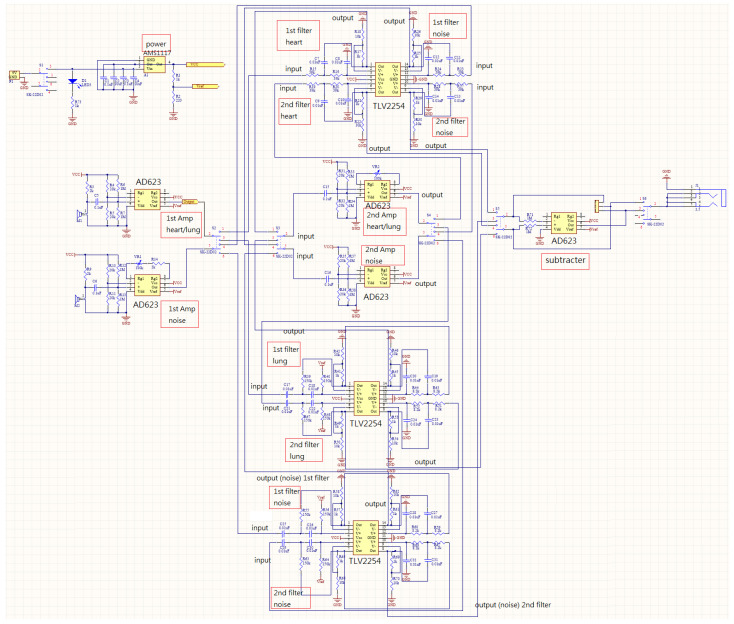
Circuit diagram of the presented system.

**Figure 7 sensors-22-04263-f007:**
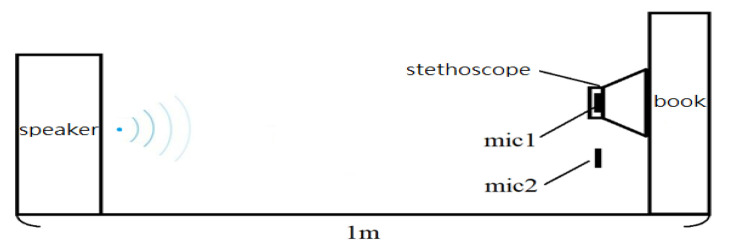
Schematic diagram of the experimental site I.

**Figure 8 sensors-22-04263-f008:**
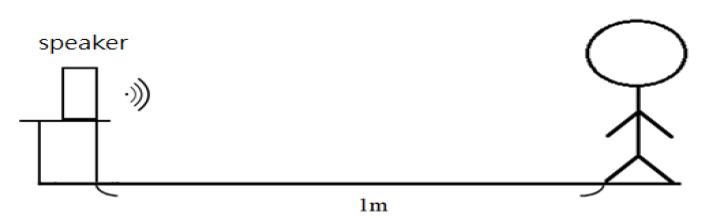
Schematic diagram of experimental site II.

**Figure 9 sensors-22-04263-f009:**
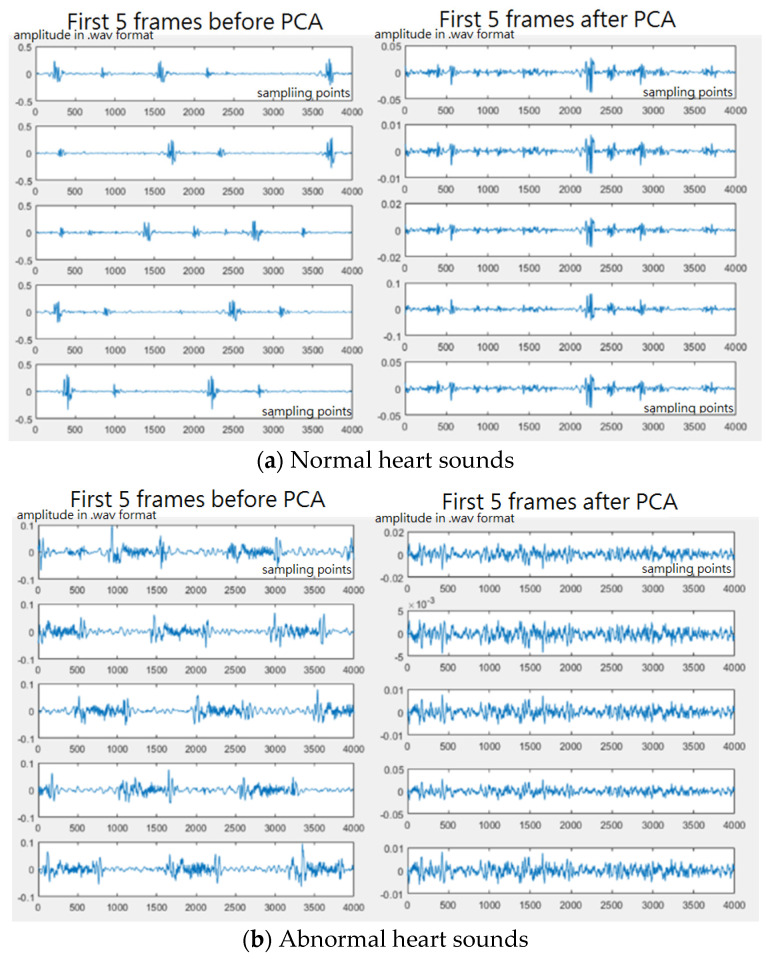
(**a**) Normal heart sounds before/after PCA and (**b**) abnormal heart sounds before/after PCA.

**Figure 10 sensors-22-04263-f010:**
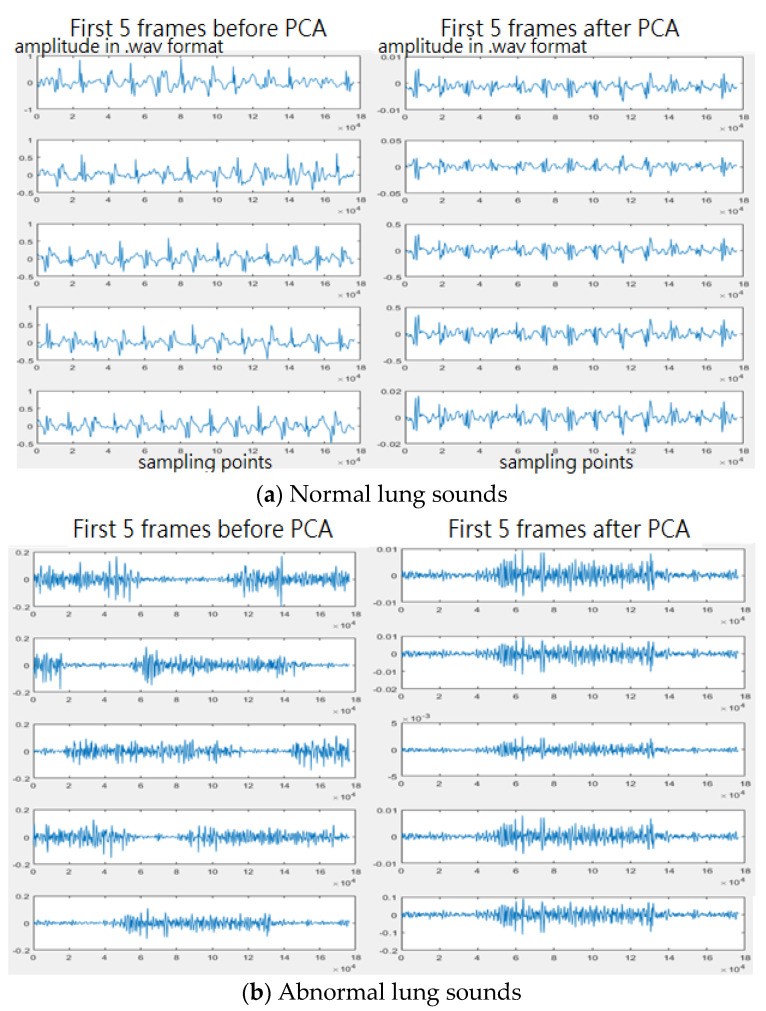
(**a**) Normal lung sounds before/after PCA and (**b**) abnormal lung sounds before/after PCA.

**Figure 11 sensors-22-04263-f011:**
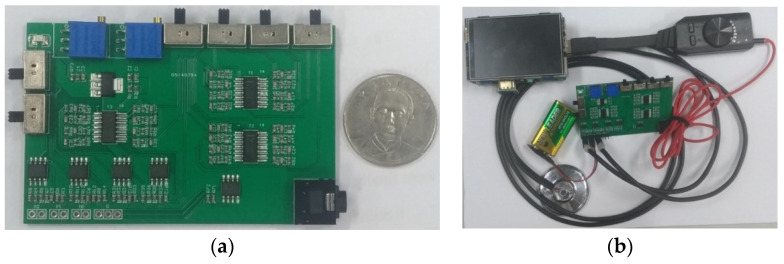
Physical PCB circuit. (**a**) PCB of designed circuit. (**b**) Whole designed system.

**Figure 12 sensors-22-04263-f012:**
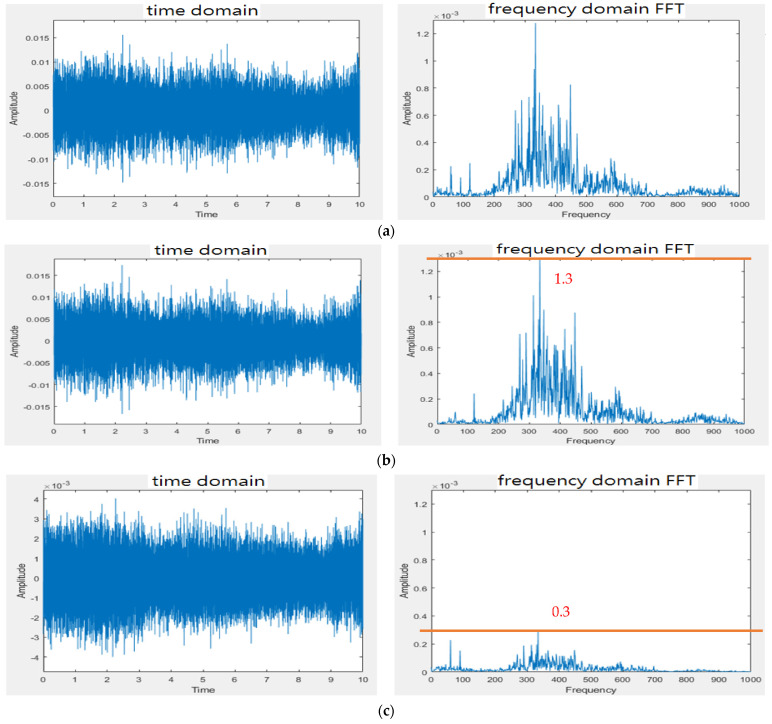
Noise reduction effect for ward construction noise. (**a**) Signal waveform and FFT spectrum measured by mic 1. (**b**) Signal waveform and FFT spectrum measured by mic 2. (**c**) Signal waveform and FFT spectrum after noise reduction.

**Figure 13 sensors-22-04263-f013:**
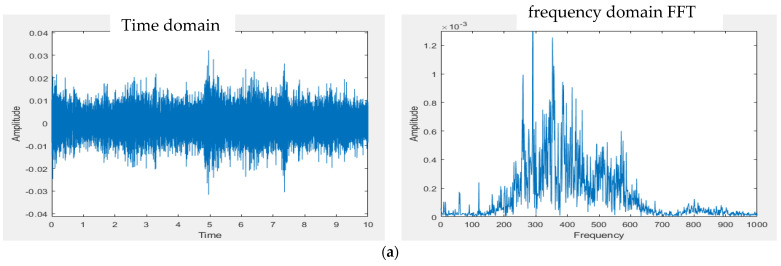
Noise reduction effect for airport noise. (**a**) Signal waveform and FFT spectrum measured by mic 1. (**b**) Signal waveform and FFT spectrum measured by mic 2. (**c**) Signal waveform and FFT spectrum after noise reduction.

**Figure 14 sensors-22-04263-f014:**
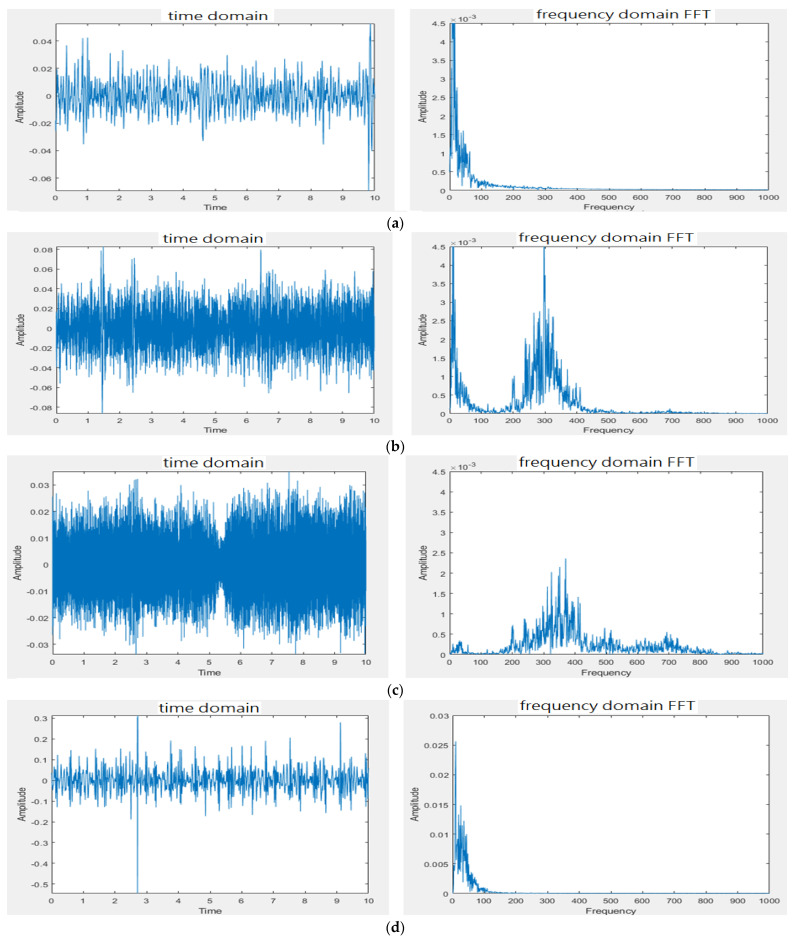
Noise reduction effect for ward construction noise. (**a**) Measurement 1: stethoscope on left leg skin in quiet space. (**b**) Measurement 2: stethoscope on the left leg skin in noise space. (**c**) Measurement 3 on lower leg skin: mic for environmental noise. (**d**) Measurement 4: stethoscope on left chest in quiet environment. (**e**) Measurement 5: stethoscope on left chest in noisy environment. (**f**) Measurement 6 on left chest after noise reduction.

**Figure 15 sensors-22-04263-f015:**
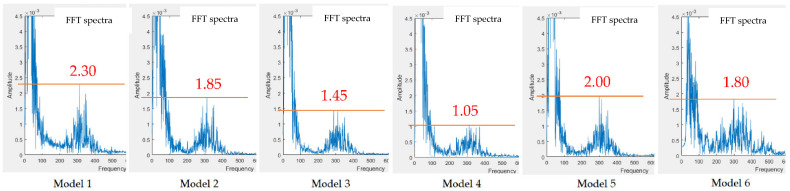
Comparison of FFT spectra of heart sound signals without using the subtracter.

**Figure 16 sensors-22-04263-f016:**
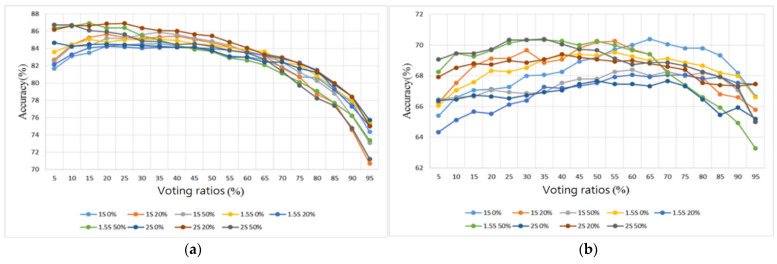
Heart sound classification results: Accuracy. (**a**) Testing samples. (**b**) Noisy samples.

**Figure 17 sensors-22-04263-f017:**
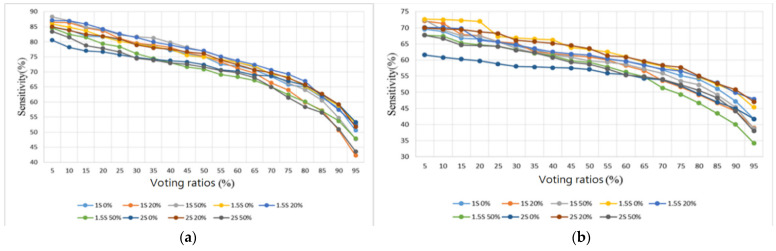
Heart sound classification results: Sensitivity. (**a**) Testing samples. (**b**) Noisy samples.

**Figure 18 sensors-22-04263-f018:**
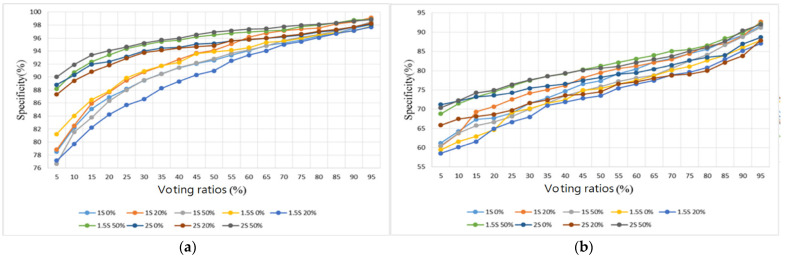
Heart sound classification results: Specificity. (**a**) Testing samples. (**b**) Noisy samples.

**Figure 19 sensors-22-04263-f019:**
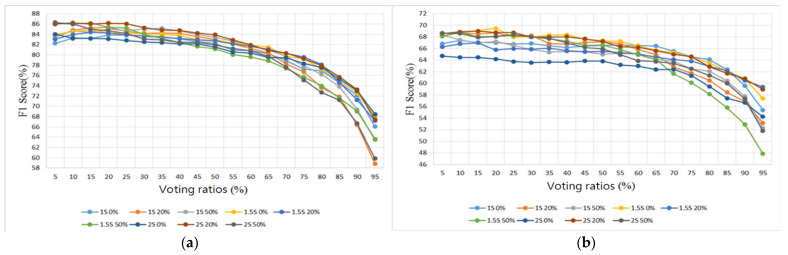
Heart sound classification results: F1 Score. (**a**) Testing samples. (**b**) Noisy samples.

**Figure 20 sensors-22-04263-f020:**
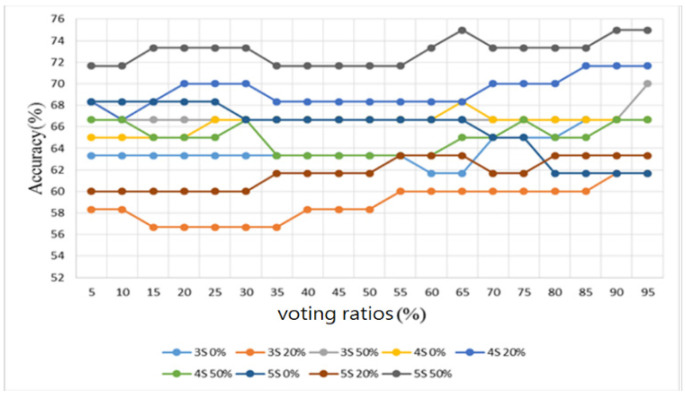
Lung sound classification results: Accuracy.

**Figure 21 sensors-22-04263-f021:**
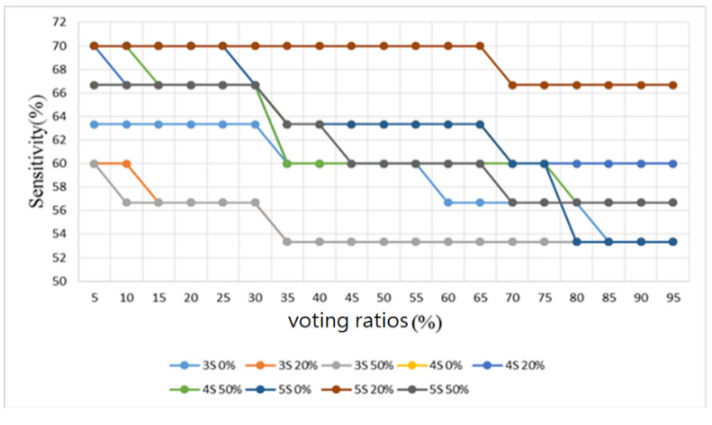
Lung sound classification results: Sensitivity.

**Figure 22 sensors-22-04263-f022:**
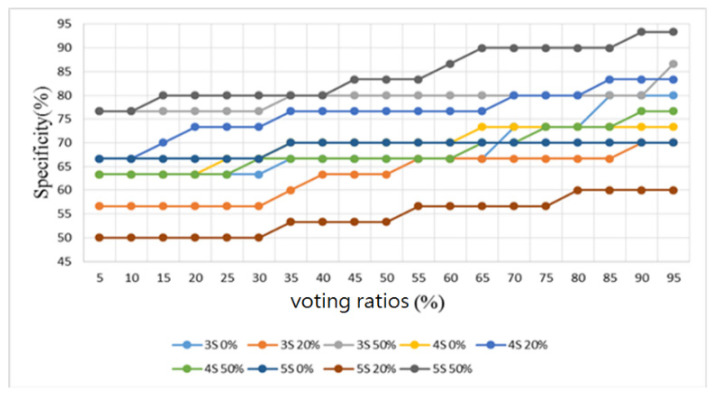
Lung sound classification results: Specificity.

**Figure 23 sensors-22-04263-f023:**
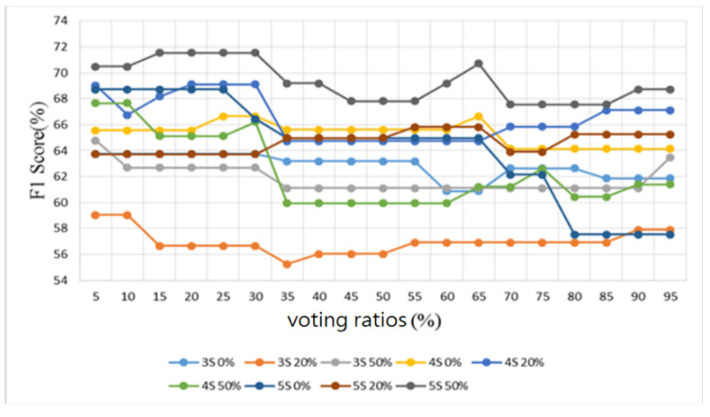
Lung sound classification results: F1 Score.

**Table 1 sensors-22-04263-t001:** Six different models of electronic stethoscope heads.

Model 1	Model 2	Model 3
One stethoscope One microphone (without cork)	One stethoscope One mirophone (with cork)	Two stethoscopes (without cork)
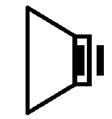	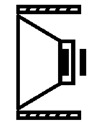	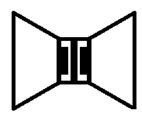
**Model 4**	**Model 5**	**Model 6**
Two stethoscope (with cork)	One stethoscope covered with cork	Thinklabs One Digital Stethoscope
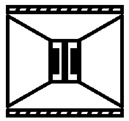	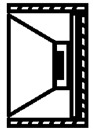	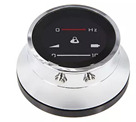

Note: Areas with slashes represent cork.

**Table 2 sensors-22-04263-t002:** Frequency bands (in Hz) of different measurements for different models.

	Model 1	Model 2	Model 3	Model 4	Model 5
Measurement 1	0~100	0~100	0~100	0~100	0~100
Measurement 2	0~100	0~100	0~100	0~100	0~100
	150~450	150~450	150~450	150~450	150~450
	peak at 300	peak at 300	peak at 300	peak at 300	peak at 300
Measurement 3	150~550	150~550	150~450	250~450	
	peak at 375	peak at 375	peak at 350	peak at 350	
Measurement 4	0~150	0~150	0~150	0~150	0~150
Measurement 5	0~150	0~150	0~150	0~150	0~150
	200~450	200~400	240~420	250~400	250~400
	peak at 325	peak at 325	peak at 310	peak at 310	peak at 310
Measurement 6	0~150	0~150	0~150	0~150	
	250~450	250~450	250~450	250~450	

**Table 3 sensors-22-04263-t003:** Comparison of FFT spectra of heart sound signals with and w/o the subtracter.

Model 1		Model 2	
w/o subtracter	with subtracter	w/o subtracter	with subtracter
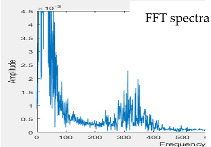	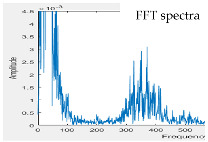	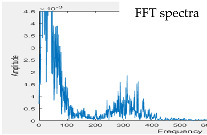	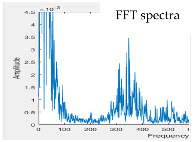
**Model 3**		**Model 4**	
w/o subtracter	with subtracter	w/o subtracter	with subtracter
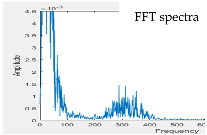	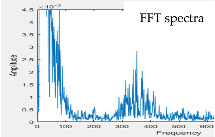	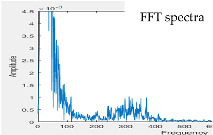	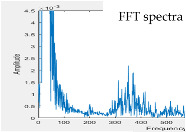

**Table 4 sensors-22-04263-t004:** Sizes of heart sound samples.

Training Samples		Testing Sampes		Noisy Samples	
Abnormal	Normal	Abnormal	Normal	Abnormal	Normal
411	1940	103	485	151	150

**Table 5 sensors-22-04263-t005:** Comparison results.

Models	Training #: Testing #	Accuracy	Sensitivity	Specificity	F1
Adaptive Boosting + CNN [17]	9:1	86.02	94.24	77.81	--
DNN [18]	9:1	97.10	99.26	94.86	--
WST + PCA + 2SVM [20] *	7:3	93.06	--	--	--
Classic ML + DL [21]	9:1	92.9	82.3	96.2	--
1D CNN+ BiLSTM [22]	9:1	86.57	91.78	59.05	91.78
Ensemble-NN [33]	9:1	91.5	94.23	88.76	--
DropConnected-NN [34]	9:1	84.1	84.8	93.3	--
Adaptive Boosting + CNN [17] **	--	89.6	93.7	85.6	90
Ensemble-NN [33] **	--	93.0	94.5	91.4	93.1
DropConnected-NN [34] **	--	93.1	94.5	91.7	93.1
Presented model	4:1	86.9	81.9	91.8	86.1

#: number(s). DNN: deep neural network. ML: machine learning. DL: deep learning. NN: neural network. CNN: convolutional neural network. WST + 2SVM: Wavelet scattering transformation (WST) + twin support vector machine (2SVM). 1D CNN + BiLSTM: 1D CNN (1DCNN) + bi-directional long short-term memory (BiLSTM). 9:1 means 10-fold evaluation approach in PhysioNet dataset. *: dataset A in PhysioNet dataset. **: The test using the program and trained weights of NN was obtained from the PhysioNet website [28]. --: data not available.

## Data Availability

Not applicable.

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
