# Peer review of "Development of an Electronic Stethoscope and a Classification Algorithm for Cardiopulmonary Soundsâ€"

_sensors, 2022, doi:10.3390/s22114263_

Round 1
Reviewer 1 Report
In this paper, an electronic stethoscope embedding a condenser microphone in the traditional stethoscope head to collect cardiopulmonary sounds was developed. The work is adequate for this journal, but the structure of the paper still has problems and needs further improvement. The comments are listed below:
1.The connection between the development of the electronic stethoscope and the classification algorithm is not clear. For example, can the data collected from the developed electronic stethoscope improve the classification accuracy or efficiency of the algorithm? Please explain it in detail.
2.Whether the improved classification algorithm is compared with the previous algorithm? Please add this part in the article.
3.The importance and the existing problems of the stethoscope should be highlighted in the abstract.
4.This paper in the introduction expounded two types of stethoscopes and some signal processing methods. Compared with their work, what are the advantages and disadvantages of this study?
Author Response
Thanks a lot for the reviewer's valuable comments. The following are responses to these comments.
1. The connection between the development of the electronic stethoscope and the classification algorithm is not clear. For example, can the data collected from the developed electronic stethoscope improve the classification accuracy or efficiency of the algorithm? Please explain it in detail.
Response:
Heart and lung sounds are important biomedical information to help diagnose heart disease and respiratory disease. With conventional stethoscopes, the auscultation results may vary from one doctor to another doctor due to the decline in his/her hearing ability with age or his/her different professional training background, and the problematic cardiopulmonary sound cannot be recorded for analysis. In addition, heart disease patients often have recurrences after the onset of the disease, so the heart sounds and electrocardiograms (ECG) of heart disease patients must be measured frequently. Such real-time cardiopulmonary sound data, if available, can provide further identification of the patient's symptoms. When an abnormal condition occurs, the patient can be alerted to contact medical staff in a timely manner, transmit ECG and cardiopulmonogram (CPG) data quickly, and return to the hospital early. The patient can be provided with appropriate medical steps on the way back to the hospital to receive proper treatment, maintain health, reduce the re-hospitalization rate, and reduce medical costs. Therefore, electronic stethoscopes that can record and analyze/classify the heart and lung sounds are important for outpatient, emergency clinical care, and telemedicine care. If an effective classification algorithm can be embedded into electronic stethoscopes, doctors can use electronic stethoscopes to obtain a prompt preliminary diagnosis in an emergency, or those who need secondary prevention after discharge from the hospital and care at home can have long-term monitoring and early detection of abnormalities. Therefore, in this study, a low-cost electronic stethoscope with noise reduction is proposed, and an effective classification algorithm, based on principal component analysis (PCA), MFCC, and Ensemble Learning, is developed for auscultatory sounds. However, in Taiwan, collecting patients’ data for analysis has to be approved by the Institutional Review Board (IRB) first. Moreover, even the approval is granted, collecting enough data is time-and-labor-consuming. Therefore, in order to expedite the advance of AI classification algorithms for auscultation and for the convenience of research, databases [26,27] are open to the public through the efforts of several medical groups and are used for this study. The developed algorithm then can be embedded into the designed electronic stethoscope and can make it beneficial to the medical care.
The above paragraph has been added to Abstract, Introduction, and Section 2.2.
2. Whether the improved classification algorithm is compared with the previous algorithm? Please add this part in the article.
Response:
For further comparison with considering the sample source, we chose the top three models in the 2016 PhysioNet/CinC Heart Sound Classification Challenge. Since the original training code of the authors participating in the 2016 competition was not available, only the final classification program for the competition was available, so it was not possible to compare the results fairly. The comparison results are shown in Table 5. The results show that the recognition of normal heart sounds by using our model is a little better than those by the other methods, but the recognition of abnormal heart sounds is a little worse. Although the overall score was about 4-7% different from the top three in the competition, the classification method they used was a very complex DNN method that required special hardware with high computing power to execute, and in addition to the higher cost and inconvenience of porting the model, these methods could not be executed on embedded systems with low computing power. For outpatient physicians, they are unable to obtain the patient's heart sound recognition results in real time to assist them in making correct judgments. On the other hand, our model is not complicated and with reasonable performance; therefore it can be imported to an embedded system, such as Raspberry Pi.
Table 5. Comparison results
Models |
Accuracy. |
Sensititvity |
Specificity |
F1. |
AdaBoost/CNN [15] |
89.6 |
93.7 |
85.6 |
90 |
Ensemble-NN [31] |
93 |
94.5 |
91.4 |
93.1 |
DropConnected-NN [32] |
93.1 |
94.5 |
91.7 |
93.1 |
Presented model |
86.9 |
81.9 |
91.8 |
86.1 |
The above contents have been added to Section 4.2.
3. The importance and the existing problems of the stethoscope should be highlighted in the abstract.
Response:
In the Abstract, highlights of the importance and the existing problems of the stethoscope are added.
“With conventional stethoscopes, the auscultation results may vary from one doctor to another doctor due to the decline in his/her hearing ability with age or his/her different professional training, and the problematic cardiopulmonary sound cannot be recorded for analysis. In this paper, to resolve the above-mentioned issues, an electronic stethoscope embedding a condenser microphone in the traditional stethoscope head to collect cardiopulmonary sounds was developed, and an AI-based classifier for cardiopulmonary sounds was proposed.”
4. This paper in the introduction expounded two types of stethoscopes and some signal processing methods. Compared with their work, what are the advantages and disadvantages of this study?
Response:
The advantages of this study are in several folds. Two condenser microphones deployed into double stethoscope heads surrounded by a cork (the presented stethoscope Model 4) presents certain noise isolation. The presented stethoscope is cost-effective and easily implemented. The condenser microphone used in the presented stethoscope can be found in the market with reasonable price, and the filter and amplification circuits are simple. The proposed classification algorithm is not complicated and with good performance; therefore it can be imported to an embedded system, such as Raspberry Pi. The disadvantage of this study is the subtracter for the stethoscope is not adaptive to cope with the frequency shifts due to different sound media associated with the stethoscope head and the microphone for the environmental noise.
Reviewer 2 Report
This article developed an electronic stethoscope using condenser micropphone based on a traditional stethoscope. Different deployments were investigated for evaluating the effect of noise reduction. Additionally, alogrithms of heart sounds and lung sounds classification were proposed and tested. The authors had very good literature study. The language is clear to read. But I have following concerns.
(1) I think the way to evaluate the performance of the electronic stethoscope needs improvement. In my mind, compresensive evaluation may be needed besides FFT. For example, the proposed stethoscope may be compared with an exisiting electronic stetheoscope. Pseudo signal-to-noise ratio ia an alternative indicator to check the effect of noise reduction in some degree. Or, a measured frequency fresponse of the stethoscope over the frequency band can show the performance too.
(2) The figures, such as figures 10-13 are not clearly for checking by human eyes. The time scale in these figures is not good for deep understanding.
(3) Why did the authors introduce the heart sounds and lung sounds from data base [25, 26]? I cann't get the authors' purpose. To my understanding, this article was to describe and evaluate the proposed stethoscope. The signals from the database were not collected by the proposed stethoscoep. This part may be put in another article if the authors wish to present the new classification algorithms.
Author Response
Thanks a lot for the reviewer's valuable comments. The following are responses to these comments.
1. I think the way to evaluate the performance of the electronic stethoscope needs improvement. In my mind, comprehensive evaluation may be needed besides FFT. For example, the proposed stethoscope may be compared with an existing electronic stethoscope. Pseudo signal-to-noise ratio is an alternative indicator to check the effect of noise reduction to some degree. Or, a measured frequency response of the stethoscope over the frequency band can show the performance too.
Response:
For experimental site I (Figure 6), two different noise sources were played. Figure 10 shows the 10-second sound waveforms and FFTs measured by mic 1, mic 2, and the subtracter (noise reduction) for ward construction noise. Based on the FFT spectrum, the noise amplitude peak before noise reduction is about 0.0013, while it is 0.0003 after noise reduction. Apparently, the volume after noise reduction is 4 times lower than it before noise reduction; the noise reduction is about 12.74 dB based on the formula for the signal-to-noise ratio (SNR). The 10-second sound waves and FFTs measured by mic 1, mic2, and noise reduction for airport noise are shown in Figure 11. For airport noise played from the computer with maximal volume and the speaker with largest volume, the noise amplitude peak before noise reduction is about 0.0013, and it is about 0.00025 after noise reduction. In this case, the volume after noise reduction is 5.2 times lower than it before noise reduction; the noise reduction is about 14.32 dB. Figure 13 also shows the cork can reduce the noise by 1.9 dB from model 1 to model 2 and by 2.8 dB from model 3 to model 4. Model 4 can reduce the noise by 4.7 dB when compared with the commercial one, model 6.
The above paragraph has been added to Section 3.1.
2. The figures, such as figures 10-13 are not clearly for checking by human eyes. The time scale in these figures is not good for deep understanding.
Response:
Figures are enlarged for a better view. The time of sound waveforms in Figures is all 10 seconds and is explicitly indicated in the text (lines 344, 350, and 377).
3. Why did the authors introduce the heart sounds and lung sounds from data base [25, 26]? I can’t get the authors' purpose. To my understanding, this article was to describe and evaluate the proposed stethoscope. The signals from the database were not collected by the proposed stethoscope. This part may be put in another article if the authors wish to present the new classification algorithms.
Response:
Heart and lung sounds are important biomedical information to help diagnose heart disease and respiratory disease. With conventional stethoscopes, the auscultation results may vary from one doctor to another doctor due to the decline in his/her hearing ability with age or his/her different professional training background, and the problematic cardiopulmonary sound cannot be recorded for analysis. In addition, heart disease patients often have recurrences after the onset of the disease, so the heart sounds and electrocardiograms (ECG) of heart disease patients must be measured frequently. Such real-time cardiopulmonary sound data, if available, can provide further identification of the patient's symptoms. When an abnormal condition occurs, the patient can be alerted to contact medical staff in a timely manner, transmit ECG and cardiopulmonogram (CPG) data quickly, and return to the hospital early. The patient can be provided with appropriate medical steps on the way back to the hospital to receive proper treatment, maintain health, reduce the re-hospitalization rate, and reduce medical costs. Therefore, electronic stethoscopes that can record and analyze/classify the heart and lung sounds are important for outpatient, emergency clinical care, and telemedicine care. If an effective classification algorithm can be embedded into electronic stethoscopes, doctors can use electronic stethoscopes to obtain a prompt preliminary diagnosis in an emergency, or those who need secondary prevention after discharge from the hospital and care at home can have long-term monitoring and early detection of abnormalities. Therefore, in this study, a low-cost electronic stethoscope with noise reduction is proposed, and an effective classification algorithm, based on principal component analysis (PCA), MFCC, and Ensemble Learning, is developed for auscultatory sounds. However, in Taiwan, collecting patients’ data for analysis has to be approved by the Institutional Review Board (IRB) first. Moreover, even the approval is granted, collecting enough data is time-and-labor-consuming. Therefore, in order to expedite the advance of AI classification algorithms for auscultation and for the convenience of research, databases [26,27] are open to the public through the efforts of several medical groups and are used for this study. The developed algorithm then can be embedded into the designed electronic stethoscope and can make it beneficial to the medical care.
The above paragraph has been added to Abstract, Introduction, and Section 2.2.
Reviewer 3 Report
Thanks for recommending me as a reviewer. In this paper, authors an electronic stethoscope embedding a condenser microphone in the traditional stethoscope head to collect cardiopulmonary sounds was developed, and an AI-based classifier for cardiopulmonary sounds was proposed. Different deployments of the microphone in the stethoscope head with amplification and filter circuits were explored and analyzed using Fast Fourier Transform (FFT) for evaluating the effect of noise reduction. In this paper, after testing, the microphone placed in the stethoscope head surrounded by cork is found to have better noise reduction. If authors complete minor revisions, the quality of the study will be further improved.
1. The introduction section is well written, but too verbose. If authors shorten the introductory section to be more clear, it can help readers understand.
2. line 369: Please correct the typo in Equation (3). ex. nsitivity -> sensitivity
3. In Table 2, the unit (e.g. Hz) is recommended to be expressed in the title.
Author Response
Thanks a lot for the reviewer's valuable comments. The following are responses to these comments.
1. The introduction section is well written but too verbose. If authors shorten the introductory section to be more clear, it can help readers understand.
Response:
The Introduction has been shortened by more than 20%.
2. line 369: Please correct the typo in Equation (3). ex. nsitivity -> sensitivity
Response:
It has been corrected.
3. In Table 2, the unit (e.g. Hz) is recommended to be expressed in the title.
Response:
It has been corrected.
Reviewer 4 Report
GENERAL COMMENT
The authors presented a study on the development of an electronic stethoscope, based on a chest-piece from a normal stethoscope instrumented with a condenser microphone, surrounded by cork to obtain a physical noise reduction. An AI-based approach was also proposed for heart and lung sounds classification.
The description of the “cardiopulmonary sound classification” in the abstract is rather vague, particularly the ultimate goal of the classification. Which was the purpose of the classification? Discriminating between heart sounds and lung sounds? Discriminating between sounds from healthy and pathological subjects? In general, I suggest that the authors re-organize the abstract to clarify the methodology and better highlight the novelty of their work.
The first part of the introduction lacks proper references to the pertaining literature, particularly lines 58-64. Moreover, I suggest that the authors enrich the description of the contact-based devices for heart/lung sounds acquisition with the recent advancements in Forcecardiography, which is a novel technique to measure the heart mechanical vibrations of the chest wall, and particularly the topics of "Forcecardiography sensors for respiration monitoring" and "Novel piezoelectric Forcecardiography sensors to measure respiration, infrasonic cardiac vibrations and heart sounds simultaneously from a single point on the chest".
SPECIFIC COMMENTS
Regarding the five different stethoscope models that were tested in the study, you must provide clear rationale behind their design and the expected variations in performances.
Concerning the pre-processing of the stethoscope signals, the application of the principal component analysis to signals frames is very obscure. First of all, what is the rationale behind this processing? What did you expect to obtain by using this methodology before extracting the features for the classification and why? Also, at lines 272-273 you referred to “dimensionality reduction” and “signal decomposition”. Can you explain in more detail what do you mean for “dimensionality reduction” and how did you achieve it, and how did you perform “signal decomposition”?
More specific comments:
- Lines 189-191: “transmitted to a Raspberry Pi” suggests a kind of digital signal transmission, but the source is a differential amplifier, so I guess you meant to say that the signals are fed to the analog inputs of a Raspberry Pi, which provided for their analog-to-digital conversion (A/D). In these regards, you must specify the actual sample rate and resolution (i.e. number of bits) adopted for the A/D conversion.
- Figure 2: in panel (b) it is not clear if a whole layer of cork was placed onto the condenser microphone before closing with the white diaphragm, or if the cork was used to make a sort of gasket to seal the space between the diaphragm and the condenser microphone. To clarify this point, please include a picture of the device with the applied cork, but without the white diaphragm.
- Lines 206-211: Why didn’t you apply external noise reduction by subtracting the external noise signal picked up by mic2 from the heart/lung sounds signal picked up by mic1 before applying any filter to extract the specific heart/lung sounds components? It seems that you wasted electronic resources by realizing heart/lung sounds filters to be applied to the noise signals (which is not useful), only for the purpose of obtaining the same time lag to eventually subtract in phase the external noise signal provided by mic2 from the stethoscope sound signal provided by mic1.
- Lines 240-241: this methodology seems not reasonable: why should a thick book be representative of the acoustic properties of the human torso? There is no reference to the scientific literature supporting the adoption of this methodology. Please explain the rationale of your choice.
- Line 241: What is a MECMAR speaker? Please specify it in the text and add a datasheet of the device to the references.
- Lines 342-373: In the whole paragraph on the classifier, you never clearly stated what was the goal of the classification, i.e. which were the classes to be discriminated by the classifier. Please clarify.
Round 2
Reviewer 1 Report
The author has well addressed the problems. I suggest the acceptance in the current version.
Author Response
Thank you very much for reviewing our manuscript.
Reviewer 2 Report
I have no further comments. The manuscript quality has improved in some degree.
Author Response

(The authors gave the same response as above.)

Reviewer 4 Report
Please see the attached file for comments.

Author Response
Thanks a lot for the reviewer's valuable comments. Please find the attached pdf file for responses to those comments.

Round 3
Reviewer 4 Report
I have only some minor comments.
1) I suggest that the authors add a table to compare the results of their classification approach with the results of the articles that were analyzed in the introduction of the manuscript. This would improve the readers' comprehension of the impact of this study in this research field.
2) Figure 6: The yellow rectangles have no labels. You must report the labels of the devices corresponding to those rectangles.
3) Figures 9 and 10: you must label x and y axes, also reporting the measurement units.
4) Lines 178-185: You must cite proper references to the literature to support these statements. The human audible range is commonly considered to be 20 Hz - 20 kHz, so stating that human ear is "sensitive in the range of 40 Hz to 400 Hz" can be confusing, all the more so without a proper reference to the literature. What do you mean exactly by "sensitive"? How has the "sensitive" range of 40 Hz - 400 Hz been identified for the human ear?
Author Response
Thanks a lot for the valuable comments. The responses to these comments are as follows.
1) I suggest that the authors add a table to compare the results of their classification approach with the results of the articles that were analyzed in the introduction of the manuscript. This would improve the readers' comprehension of the impact of this study in this research field.
Response:
Except for the comparison results of the top three models [17,33,34] in the 2016 PhysioNet/CinC Heart Sound Classification Challenge [28] and the presented model, Table 5 has been modified to incorporate the test results collected from [17,18,20-22,33-34] that were reviewed in Introduction. Lines 609-612 are added.
“The performance indices of other models [17,18,20-22,33-34] that were reviewed in Introduction are also listed in rows 1-7 of Table 5. However, since we don’t have the access to the codes of these models, Table 5 only lists the test results collected from the literature.”
Models |
Training #: Testing # |
Accuracy |
Sensitivity |
Specificity |
F1 |
Adaptive Boosting + CNN [17] |
9:1 |
86.02 |
94.24 |
77.81 |
-- |
DNN [18] |
9:1 |
97.10 |
99.26 |
94.86 |
-- |
WST+PCA+2SVM [20]* |
7:3 |
93.06 |
-- |
-- |
-- |
Classic ML+ DL [21] |
9:1 |
92.9 |
82.3 |
96.2 |
-- |
1D CNN+ BiLSTM [22] |
9:1 |
86.57 |
91.78 |
59.05 |
91.78 |
Ensemble-NN [33] |
9:1 |
91.5 |
94.23 |
88.76 |
-- |
DropConnected-NN [34] |
9:1 |
84.1 |
84.8 |
93.3 |
-- |
Adaptive Boosting + CNN [17]** |
-- |
89.6 |
93.7 |
85.6 |
90 |
Ensemble-NN [33]** |
-- |
93.0 |
94.5 |
91.4 |
93.1 |
DropConnected-NN [34]** |
-- |
93.1 |
94.5 |
91.7 |
93.1 |
Presented model |
4:1 |
86.9 |
81.9 |
91.8 |
86.1 |
DNN: deep neural network
ML: machine learning
DL: deep learning
NN: neural network
CNN: convolutional neural network
WST+2SVM: Wavelet scattering transformation(WST)+twin support vector machine(2SVM)
1D CNN+ BiLSTM: 1D CNN (1DCNN) + bi-directional long short time memory(BiLSTM)
9:1 means 10-fold evaluation approach for PhysioNet dataset
* : dataset A in PhysioNet dataset
**: Test using the program and the trained weights of NN provided from PhysioNet website [28].
-: data not available
2) Figure 6: The yellow rectangles have no labels. You must report the labels of the devices corresponding to those rectangles.
Response:
Labels have been added in Figure 6.
3) Figures 9 and 10: you must label x and y axes, also reporting the measurement units.
Response:
Labels on the x and y axes have been added in Figures 9 and 10.
(lines 299-300) “The horizontal axis represents the sampling points, and the vertical axis represents the sound amplitude in the wav format.”
4) Lines 178-185: You must cite proper references to the literature to support these statements. The human audible range is commonly considered to be 20 Hz - 20 kHz, so stating that human ear is "sensitive in the range of 40 Hz to 400 Hz" can be confusing, all the more so without a proper reference to the literature. What do you mean exactly by "sensitive"? How has the "sensitive" range of 40 Hz - 400 Hz been identified for the human ear?
Response:
The statement at lines 178-185 can be found at http://cht.a-hospital.com/w/%E5%BF%83%E9%9F%B3 (in Chinese), Reference [25] for it was added in the manuscript.